# Solubility and Decomposition of Organic Compounds in Subcritical Water

**DOI:** 10.3390/molecules28031000

**Published:** 2023-01-19

**Authors:** Erdal Yabalak, Sema Akay, Berkant Kayan, A. Murat Gizir, Yu Yang

**Affiliations:** 1Department of Nanotechnology and Advanced Materials, Mersin University, 33343 Mersin, Turkey; 2Department of Chemistry, Arts & Science Faculty, Aksaray University, 68100 Aksaray, Turkey; 3Department of Chemistry, Faculty of Science, Mersin University, 33342 Mersin, Turkey; 4Department of Chemistry, East Carolina University, Greenville, NC 27858, USA

**Keywords:** subcritical water (SBCW), solubility, decomposition, organic compounds

## Abstract

In this article, studies on organic solubility and stability in subcritical water reported during the past 25 years have been reviewed. Data on the solubility and decomposition of organic compounds in subcritical water, a green solvent, are needed in environmental remediation, chemistry, chemical engineering, medicine, polymer, food, agriculture, and many other fields. For solubility studies, the experimental systems used to measure solubility, mathematical equations derived and applied for the modeling of the experimentally determined solubility data, and the correlation between the predicated and experimental data have been summarized and discussed. This paper also reviewed organic decomposition under subcritical water conditions. In general, the solubility of organics is significantly enhanced with increasing water temperature. Likewise, the percentage of organic decomposition also increases with higher temperature.

## 1. Introduction

The use of subcritical water as a green solvent for extraction or reaction media has gained importance with advanced scientific studies in the last 20 years. Subcritical water has variable physical properties compared to water at ambient conditions such as the dielectric constant, which is typically used for measuring polarity and can easily be tuned by changing temperature and pressure. As the temperature rises above 373, 473 and 505 K, the dielectric constant of water reaches the normal values of dimethyl sulfoxide (DMSO) (46.68), acetonitrile (37.5), and methanol (32.7), respectively, as shown in Figure 1 [1,2,3]. Therefore, recent studies have demonstrated that subcritical water is successfully used as the sole medium in both extraction and chromatography, thus completely removing organic solvents in these processes.

Solubilities of organic compounds in subcritical water are important for the design and operation of process equipment. At ambient conditions, low-polarity organics have limited solubility in water, but increasing temperature increases the solubility of these non-polar organic compounds by decreasing the dielectric constant of water [4]. The change in solubility of non-polar compounds is not only related to the decrease in the dielectric constant. More complex molecular interactions can occur when a solute is added to water. Complex molecular interactions can result in different solubility trends over a wide temperature range [5]. The solubility of hydrocarbons in water, like most other liquid-phase properties, is a weak function of pressure [6]. Thomson and Snyder [7] measured the solubility of benzene in subcritical water conditions (6.99–34.575 MPa pressure range). They obtained that the effect of pressure is small but positive as the solubility of benzene increases slightly with increasing pressure. Their results [7] were confirmed by Conolly [8] that the pressure effect on the hydrocarbon solubilities is positive but even smaller. After these studies, a simple and reliable system for the determination of solubility and partitioning behavior of fuel components in subcritical water up to 523 K was developed by Yang et al. [9]. After this study, the number of papers increased rapidly about solubility studies in subcritical water. The intermolecular interactions between organic solutes and sorbent matrices under subcritical water conditions, both polar and nonpolar organics (chlorophenols, amines, *n*-alkanes, and polycyclic aromatic hydrocarbons), and five different sorbent matrices (glass beads, alumina, Florisil, silica-bonded C_18_, and polymeric XAD-4 resins) have been investigated at a pressure of 50 bar and at temperatures ranging from 50 to 250 °C by Yang et al. [10]. The purpose of this review is to compile the solubility and degradation studies of organic substances in subcritical water and to provide information to researchers working on this subject.

## 2. Solubility in Subcritical Water

A simple and reliable system for the determination of solubility and partitioning behavior of fuel components in subcritical water up to 523 K was developed by Yang et al. [9] and shown in Figure 2. In this study, the solubility of toluene increased ~23 times by increasing the temperature from ambient temperature to 473 °C, but the pressure change (from 1 to 50 bar) did not affect the solubility values in the solubility studies performed at ambient temperature. The increases in the separation of benzene, toluene, ethylbenzene, xylenes, and naphthalene from gasoline to liquid water when the temperature is increased from ambient temperature to 473 °C range from 10 times for benzene to 60 times for naphthalene. Similarly, increases in the partitioning of polycyclic aromatic hydrocarbons from diesel fuel to liquid water when the temperature was increased from ambient temperature to 523 °C ranged from 130-fold for naphthalene to 470-fold for methylnaphthalene.

After this study in 1998, Miller et al. [11] studied that the solubility of anthracene, pyrene, chrysene, perylene, and carbazole were determined at temperatures ranging from 298 to 498 K and at pressures from 30 to 60 bar in subcritical water. They estimated the solubility equation based on simplifying assumptions and empirical correlations based on data presented in this work and previous reports. The calculation of solubility at desired temperature needs only knowledge of ambient temperature solubility. Equation (1) is given below:(1)$$\ln x_{2}(T)=\left(\frac{T_{0}}{T}\right)\ln[x_{2}(T_{0})]+15{\left(\frac{T}{T_{0}}-1\right)}^{3}$$
where *x*_2_(*T*_0_) refers to the solubility of organic compounds at ambient temperature, and *x*_2_(*T*) refers to the solubility of organic compounds at a calculated temperature.

The solubilities of benzene, toluene, m-xylene, p-cymene, octane, 2,2,4-trimethylpentane (isooctane), tetrachloroethylene, 1,2-dichlorobenzene, and tetraethyltin were investigated at temperatures ranging from 298 to 473 K. Increasing the temperature by 175 K increased the solubilities by a factor of 10–250 [12].

They also claimed that refinements to the equations, perhaps including the molecular characteristics of the solute, are possible when more experimental data become available. From 1998 until the present day, over 50 papers were published, and this basic model has been developed by several research groups.

### 2.1. Solubilities of Polycyclic Aromatic Hydrocarbons and Derivatives in Subcritical Water

The solubility of PAHs is important for many industrial plants. Furthermore, their aqueous solubility determines (Table 1) both their uptake by the roots of plants and their transfer to other parts of the plant and their mobility in the soil. The solubilities of three PAHs, namely acenaphthene, anthracene, and pyrene, in water were measured in temperature and pressure ranges of 323–573 K and 50–100 bar, respectively, by Andersson et al. [13]. The solubility values of the employed compounds below their melting point were determined to be consistent with literature values, and the solubility of pyrene and anthracene exponentially varies with temperature. The solubilities of acenaphthene, anthracene, and pyrene were calculated as mole fraction solubilities (*x*_2_) and were determined as 1.25 × 10^−3^ at 300 K and 100 bar, within 1.02 × 10^−7^–3.78 × 10^−3^ at a temperature range of 373–573 K and pressure of 50 bar and 6.87 × 10^−8^–1.41 × 10^−3^ at a temperature range of 323–573 and pressure range of 50–100 bar, respectively (Table 1).

Karásek et al. developed a semiempirical relationship to correlate the solubility of PAHs (naphthalene, anthracene, pyrene, chrysene, 1,2-benzanthracene, triphenylene, perylene, *p*-terphenyl) in pressurized hot water within the temperature range of 313–498 K, a pressure of 1–77 bars and equilibrium mole fraction (*x*_2_) of 10^−11^–10^−3^. They used only pure-component properties such as cohesive energy density, internal pressure and dielectric constant of water and enthalpy of fusion, triple-point temperature, the molar volume of the solid compound and the molar volume of the subcooled liquid of PAHs [14]. The *x*_2_ data were experimental values of the previously reported research studies. *γ*_2_ (Raoult’s law activity coefficient of the solute) values of each PAH mentioned above were calculated using Equation (3), where $f_{2}^{s0}$(the fugacity of the pure solid solute) and $f_{2}^{10}$(the pure subcooled liquid solute) values were calculated by Equations (2) and (3).
(2)$$x_{2}=\frac{f_{2}^{s0}}{\gamma_{2}^{}f_{2}^{10}}$$
(3)$$\ln\frac{f_{2}^{s0}}{f_{2}^{10}}\approx \frac{\Delta h_{2}^{\mathrm{fus}}}{RT_{t2}}\left(1-\frac{T_{t2}}{T}\right)+\frac{\left(v_{2}^{s0}-v_{2}^{10}\right)P}{RT}$$

Karásek et al. In addition measured the solubility of naphthalene (two-ring PAH), anthracene, *p*-terphenyl (three-ring PAHs), 1,2-benzanthracene and triphenylene (four-ring PAHs) in pressurized hot water at a temperature range of 313 K–melting point and pressure range of 40 to 80 bar (mp indicates of the melting point of the related compound) by a dynamic method combined with GC-MS [15]. The apparatus used in this study employs a capillary restrictor to minimize the system volume downstream of the sampling point. Although the solubility values obtained in this study were found to be compatible with those reported in the literature, it is the first time that the solubilities of the other three compounds are reported in the presented study. They evaluated the effect of temperature and hydrocarbon structure on the solubilities, namely the equilibrium mole fractions (*x*_2_) of the employed PAHs. They reported *x*_2_ values of naphthalene, between 8.49 × 10^−10^ and 2.2 × 10^−4^ at a temperature range of 313.2–483.2 K and pressure range of 49–77 bar using Equation (3) (Table 1). In addition, it was indicated that the difference in the solubility curvature of the plots, which were obtained for the employed PAHs, may be due to the structural difference among the PAHs.

The aqueous solubilities (*x*_2_) of solid heterocyclic analogues of anthracene, phenanthrene and fluorene at a specific temperature range (313 K–the melting point of each compound) under 50 bar of pressure were reported by Karásek et al. [16]. They collected the solubility data of each compound via the dynamic saturation method based on pressurized hot water extraction. *x*_2_ values for the employed compounds were found to be within the 3.17 × 10^−9^–8.27 × 10^−4^ range and were widely changeable based on the applied temperature. It was also indicated that no appreciable degradation was observed for any compound in the temperature range studied based on the GC/MS results. Obtained solubilities were converted to activity coefficients of individual solutes in saturated aqueous solutions, and the relationship between temperature and type or number of heteroatoms was evaluated (Equation (4)).
(4)$$\ln x_{2}=b_{1}+b_{2}\left(\frac{T_{0}}{T}\right)+b_{3}\ln(\frac{T}{T_{0}})$$
where *T*_0_ and *γ*_2_ refer to 298.15 K and Raoult’s law activity coefficient of the related compound, respectively. *b*_1_, *b*_2_, and *b*_3_ denote the least-squares estimates of the coefficients, and *T* is the absolute temperature at the experimental conditions. Hence, the increase in the aqueous solubilities of solid heterocyclic analogues of anthracene, phenanthrene and fluorine was reported to strongly depend on the increasing temperature and variance with the heteroatoms.

Fornari et al. applied three thermodynamic models (UNIFAC, modified-UNIFAC and A-UNIFAC) to predict the solubility of polycyclic aromatic hydrocarbons (PAHs) in subcritical water as a function of temperature (298–500 K) [17]. The experimental data for the studied compounds were collected from the previously studied literature. The modified-UNIFAC model provided the best solubility results where the solubility of hydrophobic organic compounds increased with a decrease in the dielectric constant of the subcritical water according to the A-UNIFAC model.

The solubility values of PAHs in subcritical water were calculated using Equations (5) and (6) with the above-mentioned three UNIFAC-based thermodynamic models.
(5)$$\ln x_{2}^{\mathrm{id}}=\ln\left(\frac{f_{2}^{s}}{f_{2}^{o}}\right)=-\frac{\Delta H_{m2}}{RT_{m2}}\left(\frac{T_{m2}}{T}-1\right)$$
(6)$$\ln x_{2}=\ln x_{2}^{\mathrm{id}}-\ln\gamma_{2}$$
where $f_{2}^{s}$, $f_{2}^{o}$, $\gamma_{2}$, $x_{2}$, $T_{m2}$ and $\Delta H_{m2}$ indicate the fugacity of the pure solid solute, the fugacity of the pure solute in liquid, the activity coefficient of the solute in the liquid state, molar fraction (solubility) of the related solid, the normal melting temperature of the solid and the enthalpy of the fusion, respectively. In addition, the subscript indicates the ideal conditions in which the activity coefficient of the solute in the liquid state (*γ*_2_) is equal to 1. In this case, the solubility of PAHs depends on the melting properties and temperatures of the relevant PAHs.

Karásek et al. measured the aqueous solubilities of triptycene, 9-phenylanthracene, 9,10-dimethylanthracene, and 2-methylanthracene in pressurized hot water (from 313 K to the melting point of the related compound and at ≈50 bar of pressure) using a dynamic method with a flow-through extraction cell [18]. It was determined that the temperature dependence of the solubility curvature of triptycene was significantly different from those for other solutes. Furthermore, the activity coefficients of triptycene in saturated aqueous solutions were estimated from the solubility measured using the ∆C*_P_*_2_ values (pure solute heat capacity difference) by two different approaches. Another important point to mention is that the obtained solubility value of triptycene was lower than the solubility value of anthracene under every tested condition at the same temperature. Aqueous solubilities of the solute compounds (*x*_2_) along with the applied temperature range, pressure and approximation model are given in Table 1. Equation (11), which was previously used by Karásek et al. [9], was used to estimate the *x*_2_ values. Equation (12), which was reported in the same study, was also applied to estimate the activity coefficient of the employed compounds.

The prediction of the solubility of 25 PAHs was investigated using the cubic-plus-association equation of state (CPA EoS, Equation (7)) by Oliveira et al. [19]. They collected the experimental data for all studied compounds in the specific temperature (313.15–498.15 K) and pressure range (40–65 bar) from the previously studied literature. In their study, vapor pressures and liquid densities were estimated with deviation values below 1.1% and 1.4%, respectively, using the CPA model. They also stated that the solubility values can be correlated within a 6% deviation when considering the dissolution between a self-assembled molecule and non-self-assembled molecule. Although the applied model provides a 20% global average deviation for solubilities of PAHs in pressurized hot water, the prediction capability of CPA EoS is quite good when used for PAHs without liquid density data.
(7)$$x_{s}=\frac{\varphi_{s}^{L0}}{\varphi_{s}^{L}}\exp\left[-\frac{\Delta_{\mathrm{fus}}H}{R}\left(\frac{1}{T}-\frac{1}{T_{m}}\right)\right]$$
where ∆_fus_*H* indicates solute enthalpy of fusion, *T* indicates absolute temperature, *T_m_* indicates melting temperature of a solute, R refers to gas constant, *φ* indicates the fugacity coefficient (“0” indicates a pure component), and *x_s_* indicates mole fraction aqueous solubility.

Karásek et al. measured the aqueous solubilities of oxygen-containing tricyclic aromatic solids (xanthene, anthrone, xanthone, thioxanthone, 9,10-anthraquinone, and 9,10-phenanthrenequinone) under the same temperature (from 313 K to the melting point of each compound) and pressure conditions using the dynamic saturation method with a flow-through extraction cell in another study [20]. Experimental solubility values were used to estimate the activity coefficients of solutes in saturated aqueous solutions. The solubility of a solid solute (*x*_2_) was calculated using Equations (8) and (9). *x*_2_ values for the compounds were found to be within the 7.25 × 10^−8^–1.83 × 10^−3^ range.

The Raoult’s law activity coefficient of the solute in the saturated solution (*γ*_2_^sat^) can also be obtained from Equation (9), where *f*_2_^s0^ and *f*_2_^l0^ indicate the fugacity of the pure solid solute and the pure subcooled liquid solute, respectively.
(8)$$\ln x_{2}=a_{1}+a_{2}\left(\frac{T_{0}}{T}\right)+a_{3}\ln(\frac{T}{T_{0}})$$
(9)$$x_{2}=\frac{f_{2}^{s0}}{\gamma_{2}^{sat}f_{2}^{10}}$$

It was reported that the presence of oxygen atoms in the solute molecule causes the solute to be hydrophobic compared to anthracene at certain temperatures.

Teoh et al. [21] investigated the binary and triple solubility of anthracene and *p*-terphenyl in subcritical water at two different pressures (50–150 bar) and between 393 and 473 K by using a static analytical equilibrium method. They indicated the temperature as the most important effect on the solubility of polycyclic aromatic hydrocarbons (PAHs) in SBWC. When comparing the effect of pressure and temperature on solubility, the effect of pressure is relatively insignificant. They determined that the solubility of PAHs is affected primarily by the sublimation pressure and secondarily by the dielectric constant of subcritical water. They used the Peng–Robinson equation of state to correlate the aqueous solubility of PAHs under subcritical conditions, and a good agreement was obtained between the experimental and calculated values obtained for the binary systems. To reduce the degradation of the anthracene and *p*-terphenyl, the water was degassed before the experiment, and the amount of the substance was increased into the equilibrium vessel. According to the results of the FT-IR spectra, no degradation was observed in either substance. Literature searches on the stability of PAHs in subcritical water show conflicting results. Analysis of PAHs extracted at temperatures between 40 and 483 °C showed no visible degradation in the solubility study by Karasek et al. [14]. However, in a study by Yang and Hildebrand [22], some of the phenanthrene in subcritical water was reduced to several organic compounds, including phenol, naphthalene, and benzoic acid. Degradation and reduced recovery of PAHs extracted with water at high temperatures and pressures have also been observed.

Teoh et al. [23] investigated the solubility of anthracene and *p*-terphenyl in subcritical ethanol and water at two different pressures (50–150 bar) and between 393 and 473 K by using a static analytical equilibrium method (Table 1). They indicated that the ethanol composition and temperature are effective on the solubility of polyaromatic hydrocarbons in the subcritical solvent mixture. The solubilities of PAHs in subcritical ethanol increased exponentially with temperature. A new empirical approach model has been proposed to correlate the ethanol mole fraction and temperature with the solubility of anthracene and *p*-terphenyl (Equation (10)).
(10)$$\ln\;x_{\mathrm{solute}}\;=pT+qf+r\;$$
where *x*_solute_ is the solubility of the solute, *T* is the absolute temperature, *f* is the fraction of ethanol in water, and *p*, *q*, and *r* are constants. The UNIQUAC, O-UNIFAC, and M-UNIFAC models were compared, and it was found that the UNIQUAC model showed better agreement with the experimental results. The UNIQUAC model provides a good representation of the solubility of anthracene and *p*-terphenyl in ternary systems. However, in ternary systems, all models show increasing deviations from the experimental data when the ethanol concentration increases in the mixture.

### 2.2. Alkyl and Chlorobenzene Solubilities in Subcritical Water

The solubility of chlorobenzenes, which are used as intermediates in industrial products such as dyestuffs, in water, as well as their distribution rates in water and other organic solvents, is necessary to determine their distribution rates in aquatic environments or whether they tend to accumulate (Table 2). The solubility of ethylbenzene, m-xylene, and benzene in water was determined using a laboratory-made system at temperatures ranging from 298 to 473 K and a pressure of 50 bars by Mathis et al. [24]. The solubility of alkylbenzenes increased by at least one order of magnitude by increasing the temperature from 298 to 473 K. A simple and reliable approximation model was developed, Equation (11), to predict the solubility of liquid organics in subcritical water.
(11)$$\ln x_{2}(T)=\left(\frac{T_{0}}{T}\right)\ln x_{2}(T_{0})+2{\left(\frac{T-T_{0}}{T_{0}}-1\right)}^{3}$$

The solubility of chlorobenzene was measured in subcritical water using a fused silica capillary reactor (FSCR) by Pan et al. [25]. The solubility of chlorobenzene was determined to increase linearly with temperature, and the solubility was found to be 43.50 mg/g at 446 K and 71.40 mg/g at 540 K.

It has been anticipated that the solubility of many organic and inorganic compounds can be determined in subcritical media by using the FSCR technique.

Another FSCR technique was used by Bei et al. [26] to identify 4-chlorobenzene solubilities visually using a microscope and Raman Spectroscopy. The solubility of 4-chlorotoluene linearly increased nearly three-fold with increasing temperature in the range 535.45–566.95 K

### 2.3. Organic Acid Solubilities in Subcritical Water

Organic acids are used in many biomedical applications as well as in various industrial productions. In addition to many physicochemical properties, their solubility value is also an important parameter in these processes. Kayan et al. [27] investigated (Table 3) the solubility of benzoic and salicylic acids at constant pressure and different temperatures. The mole fraction solubility of benzoic acid varied from 2.22 × 10^−3^ at 298 K to 1.36 × 10^−2^ at 473 K and that of salicylic acid varied from 4.69 × 10^−5^ at 298 K to 1.02 × 10^−1^ at 473 K. The solubility of both compounds increased two-fold with increasing the temperature from 298 to 473 K. However, it was determined that benzoic acid is as stable as 473 K, but salicylic acid is partially degraded. They proposed a new approximation model to estimate the solubility of both compounds as shown in Equation (12):(12)$$\ln x_{2}(T)=(1.85\frac{T_{0}}{T}-1)\ln x_{2}(T_{0})$$
where the mole fraction solubility at any temperature *T* is *x_2_(T)*, and the ambient mole fraction solubility is given by *x_2_(T_0_)*. This model provides a better approximation of the solubility of both acids than previous models.

Srinivas et al. [28] investigated the solubility of phenolic compounds such as gallic acid hydrate, protocatechic acid and (+)-catechin hydrate between 298 and 415 K using a dynamic flow apparatus.

The aqueous solubility of gallic acid hydrate was found to vary between 1.24 × 10^−3^ at 298.75 K and 2.33 × 10^−1^ at 415.85 K. The mole fraction solubility of protocatechuic acid at the same temperatures varied between 3.55 × 10^−3^ and 1.26 × 10^−1^, respectively. The solubility of the selected phenolic compounds was measured as a function of temperature using a dynamic flow apparatus. It was found that the solubility of these compounds increased considerably with temperature. The obtained data are in agreement with the literature. Solubility data were fitted with other semi-empirical equations as well as with empirical equations such as the modified Apelblat equation to predict the water solubility of phenolic compounds for which a solubility value is known at room temperature conditions. Thermodynamic data were obtained from solubility data as a function of temperature. The obtained thermodynamic data showed that the dissolution process of phenolic compounds in water is endergonic, exothermic and entropy-driven. The solubility of phenolic compounds was approximated using the following Equation (13). The approximation model was derived as a function of temperature.
(13)$$\ln x_{2}=\left(\frac{T_{0}}{T}\right)\ln x_{2}\left(T_{0}\right)+11\left(1-\frac{T_{0}}{T}\right)$$
where *x_s_(T)* and *x_s_(T_0_)* are the mole fraction solubilities of the phenolic compounds at temperature *T* and reference temperature *T_0_*, respectively.

Terephthalic acid (TPA) has industrial importance because of its use as a raw material in polyesters. Takebayashi et al. [29] investigated the solubility of TPA in subcritical water at a constant pressure of 10 MPa and a temperature range of 349–547 K. The solubility of TPA was found to vary between 1.25 × 10^−5^ at 349 K and 2.99 × 10^−2^ at 547 K. This shows that there is an exponential increase with temperature. They expressed the temperature dependence of ln*x_2_* as Equation (14):(14)$$\ln x_{2}=a+b\left(\frac{T}{K}\right)$$

The coefficients *a* and *b* were determined by a least-squares fit to the experimental data.

Sebacic acid is known as the most biodegradable plastic monomer, which has a wide range of uses in biomedical applications. Yabalak et al. [30] investigated the solubility of sebacic acid in subcritical water, and they used the surface response method to optimize the experiments. The mole fraction solubility of sebacic acid was found to vary between 2.22 × 10^−5^ at 313 K and 25.69 × 10^−3^ at 433 K. The solubility model proposed by Kayan et al. [26] for organic acids gave the best results in approximating the solubility of sebacic acid.

### 2.4. Pharmaceutical Compounds Solubilities in Subcritical Water

Most studies have been conducted on the pharmaceutical solubility of compounds in subcritical water (Table 4). Srinivas et al. [31] applied the Hansen three-dimensional solubility parameter concept, a group contribution method, to test its prediction performance. They used the data from literature on the extraction of betulin (an antiviral agent), niacin (vitamin B3) and flavonoids (malvidin-3,O-glucoside, malvidin-3,5-diglucoside, malvidin-3,O-(6,O-p,acetyl) glucoside, malvidin-3,O-(6,O-p,coumaroyl)glucoside and catechin) from natural sources using subcritical fluid solvents such as water and ethanol [30]. They characterized and quantifies solute–subcritical solvent interactions and miscibility as a function of temperature by Hansen solubility spheres based on relative energy differences (RED). Equation (6) was used to predict and optimize the temperature and solvent conditions for extraction of the organic compounds from natural sources based on the RED values and *R_a_* value; the term in Equation (15) is calculated by Equation (16).
(15)$$RED=\frac{R_{a}}{R_{o}}$$
(16)$$R_{a}^{2}=4{\left(\delta_{D_{1}}-\delta_{D_{2}}\right)}^{2}+{\left(\delta_{P_{1}}-\delta_{P_{2}}\right)}^{2}+{\left(\delta_{H_{1}}-\delta_{H_{2}}\right)}^{2}$$
where $R_{o}$, $R_{a}^{}$, $\delta_{D}$, $\delta_{P}$ and $\delta_{H}$ indicate the radius of the Hansen sphere, the distance between the solute or solvent and the mass center of the Hansen sphere, dispersion solubility parameter, polar solubility parameter and hydrogen bonding solubility parameter, respectively, each with MPa^1/2^ unit.

Carr et al. [32] investigated the solubility of budesonide in pure and alcohol–subcritical water mixtures in the temperature range between 298 and 473 K. The methanol and ethanol were chosen as co-solvents, and in the presence of ethanol, the solubility of budesonide increased 10-fold in SBCW conditions (Table 4). According to the literature, the solubility of budesonide was directly dependent on the dielectric constant of the solvent mixture. A correlation was developed between the budesonide solubility data and the dielectric constant of SBCW.

The solubility of budesonide in the subcritical water can be estimated within 5% using the Fornari-modified Universal Functional Activity Coefficient (MF-UNIFAC) model in the range of 298 to 473 K. Multiple hydroxide side groups in budesonide may have a complex interaction with SBCW. Furthermore, budesonide was determined to be stable up to 473 K as a result of the Fourier-transform infrared (FT-IR) analysis.

Carr et al. [33] determined the subcritical water solubility of the antifungal drug griseofulvin molecule at 70 bar and in the range of 403 to 443 K. The molar fraction solubility of griseofulvin ranged from 1.60 × 10^−4^ to 5.28 × 10^−4^ (Table 4). The solubility of griseofulvin in the subcritical water can be estimated within 4% using the MF-UNIFAC model in the range of 298 to 473 K. To understand particle morphology, the solution of griseofulvin dissolved in subcritical water was injected into water at room temperature to cause rapid precipitation. This morphology shows a significant dependence on the temperature and concentration of the subcritical water.

The solubility of naproxen in the temperature range between 403 and 443 K was investigated by Carr et al. [34], and the solubility of naproxen was correlated to temperature using a M-UNIFAC model. The molar fraction solubility of naproxen is shown in Table 4. Micronization of naproxen was carried out using the adjustable solvent power of subcritical water. Furthermore, two precipitation techniques were developed. The concentration of naproxen in subcritical water affects the size of the precipitate. The size of the naproxen precipitate in a saturated solution at 443 K is one-tenth of the size of the precipitate at 403 K.

Emire et al. [35] investigated the subcritical water solubility of paracetamol, known as an anti-inflammatory drug, at constant pressure in the temperature range of 293–403 K. The mole fraction solubility of paracetamol was found to vary depending on temperature and is shown in Table 4. They proposed a new approach model, shown in Equation (17), to determine the subcritical water solubility of paracetamol.
(17)$$\ln x_{2}=\left(\frac{T_{0}}{T}\right)\ln x_{2}\left(T_{0}\right)+\frac{8}{5}\left(\frac{T-T_{0}}{T_{0}}\right)$$

The 5-Fluorouracil (5-FLU) compound is used in many cancer treatments. Akay et al. [36] investigated the solubility of the 5-FLU compound in subcritical water under a constant pressure of 5.1 MPa and in the range of 298–473 K. According to the obtained results, the solubility increased by a factor of 12 as the temperature increased from 298 to 473 K (Table 4). In addition, a new approximation model (Equation (18)) was developed to predict the solubility of 5-FLU at high temperatures and was compared with experimental data. It is known that the dielectric constant of water depends on temperature. Therefore, the new approximation model is derived to include the dielectric constant (*ε*). In addition, the compatibility with the experimental results was checked using the Apelblat equation.
(18)$$\;\ln x_{2}=\left(\frac{T_{0}}{T}\right)\ln x_{2}\left(T_{0}\right)+\frac{\varepsilon_{\left(T\right)}}{\varepsilon_{\left(T_{0}\right)}}{\left[\left(\frac{T-T_{0}}{T_{0}}\right)\right]}^{2}$$

Ciprofloxacin (CIP) is frequently used in the treatment of most bacterial infections. Pu et al. [37] measured the subcritical water solubility of ciprofloxacin using an ethanol–water mixture at constant pressure in the range of 373–443 K. The water–ethanol ratio was determined as 0%, 5% and 20% by weight. The solubility increases with increasing temperature, and the obtained solubility data were correlated to the Apelblat equation. In addition, the nanocrystallization process of CIP was carried out using polyvinylpyrrolidone (PVP) and polyethylene glycol (PEG) as stabilizers. The dissolution process of CIP nanoparticles was improved compared to the CIP raw and therefore allowed for the production of CIP nanoparticles by controlling their morphology.

The solubility of bicalutamide, megestrol acetate, prednisolone, beclomethasone dipropionate, and clarithromycin in subcritical water (SBCW) at a temperature range from 383.15 to 443.15 K and a constant pressure of 5.5 MPa was studied by Pu et al. [38]. The solubility of compounds increased exponentially by increasing temperature, and the stabilities of the solutes during the SBCW process were first investigated by FTIR spectra analysis. The solubility data also were correlated to the Apelblat equation.

Akay et al. [39] investigated the solubility of sulfadiazine, which has very low solubility in water, at constant pressure and at a temperature range of 298–403 K. The solubility of sulfadiazine in water, water + ethanol, and water + propanol mixtures was measured in this research. The mole fraction solubility of sulfadiazine in pure water was changed from 4.3 × 10^−5^ at 298 K to 63 × 10^−5^ at 403 K. The mole fraction solubility of sulfadiazine in water + ethanol mixtures at 343 K was increased from 5.6 × 10^−5^ in the mixture of 2.5% ethanol in water to 22.5 × 10^−5^ in the mixture of 20% ethanol. In the same way, the mole fraction solubility of sulfadiazine in water + propanol mixtures at 343 K is also improved with a higher percentage of propanol in water, from 6.5 × 10^−5^ in the mixture of 2.5% propanol in water to 26.5 × 10^−5^ in the mixture of 20% propanol. They used the modified Apelblat equation to acceptably predict the solubility of sulfadiazine in water and water–alcohol mixtures at different temperatures.

Escitalopram (ESC) has very low water solubility and is used in the treatment of depressive disorders. In this study, Akay et al. [40] investigated the mole fraction solubility of ESC at constant pressure and a temperature range of 298–473 K (Table 4). Based on experimental data, a new mathematical model was developed as shown in Equation (19):(19) lnx2=(T0T)lnx2(T0)+(ε(T0)ε(T))12

The thermal stability of ESC in subcritical water was investigated using thermogravimetric and differential scanning calorimeter thermograms, X-ray diffraction (XRD) spectra, and FT-IR spectra. The ESC could be stable in subcritical water at temperatures up to 473 K.

Fluconazole is an antifungal drug, and its solubility in subcritical water was first investigated by Akay et al. [41]. The solubility experiments were carried out at constant pressure and a temperature range of 298–473 K. As can be seen in Table 4, fluconazole mole fraction solubility increases 146-fold when the temperature increases to 473 K. A new approximation model (Equation (20)) was developed as shown in the following equation and was used to predict fluconazole solubility at high temperatures. The dielectric constant changes with temperature, and it is effective on that solubility. Therefore, they added the dielectric constant in the new model.
(20)$$\ln x_{2}\left(T\right)=\left(\frac{T_{0}}{T}\right)\ln x_{2\;}\left(T_{0}\right)+\ln\left(\frac{\varepsilon_{T_{0}}}{\varepsilon_{T}}+2\right)$$

In addition, a modified Apelblat equation was used to correlate the temperature dependence of fluconazole solubility in subcritical water. When the FT-IR spectra of fluconazole at 298 and 473 K were examined, they indicated that fluconazole was stable at high temperatures.

Ibuprofen (IBP) belongs to the class of pain relievers and is widely used throughout the world. Akay et al. [42] investigated the mole fraction solubility of IBP at constant pressure and a temperature range of 298–473 K. They indicated that the mole fraction solubility of IBP increases 10.600-fold when the temperature increases to 473 K. This high increase in solubility is suggested due to the strong interaction of hydrogen bonds between the water and functional groups of IBP. At subcritical conditions, the number and strength of H-bonds between water molecules are reduced; thus, more free-moving, individual water molecules exist and are available to form H-bonds with the oxygenated groups of IBP. As can be seen in Equation (21), to estimate the solubility of IBP at high temperatures, they derived a new equation that included the dielectric constant. In addition, a modified Apelblat equation was used to correlate the temperature dependence of IBP solubility in subcritical water.
(21)lnx2(T)=(1.85T0T−1)lnx2(T0)+(ε(T)ε(T0))TT0

According to the thermal analysis data, IBP shows good stability up to 398 K. Thermal decomposition occurs in the range of 398–523 K.

The solubility of ampicillin (AMP) at constant pressure in the temperature range of 303.15 and 403.15 K was studied by Mohammedi et al. [43] Mole fraction solubility is shown in Table 4 at the examined temperatures. A response surface methodology was used to understand the effect of parameters on the production of ampicillin nanoparticles, such as subcritical water temperature, polyethylene glycol concentration, and anti-solvent temperature. The analytical results confirmed that the AMP particles were nano-sized to the smallest mean size of 66.5 nm. In this study, the order of magnitude of the studied operational parameters affecting particle size could be classified as subcritical water temperature > PEG concentration > anti-solvent temperature.

Mohammadi et al. [44] examined the solubility of amiodarone hydrochloride (AMD), an antiarrhythmic drug, in pure and ethanol-modified subcritical water. The solubility experiments were carried out at temperatures ranging between 298.15 and 393.15 K and 0–10% (*w*/*w*) of ethanol as a cosolvent under constant pressure by applying a static method. The mole fraction solubility of amiodarone was calculated between 0.14 × 10^−4^ and 2.85 × 10^−4^ in the pure solvent while ranging from 0.31 × 10^−4^ to 9.82 × 10^−4^ in the ethanol-modified subcritical water. Results from FT-IR spectroscopy demonstrated the thermal stability of AMD in solution up to 393.15 K. The results also showed that the chemical structure of AMD was preserved at temperatures between 298.15 and 393.15 K.

Letrozole (LTZ) is a selective nonsteroidal aromatase inhibitor, which reduces estrogen levels produced by the body. The solubility of letrozole in subcritical water was determined at different temperatures (298.15–383.15 K) at constant pressure by applying a static method by Mohammedi et al. [45]. The experimental model Box–Behnken was used in technical analysis for the optimization of process parameters and the modeling of their relationships. They studied the influence of the parameters of the procedure, including subcritical water temperature, polyethylene glycol concentration and anti-solvent temperature, on the size and morphology of the precipitated nanoparticles was examined. The mole fraction solubility changes were approximately 23-fold in the studied temperatures (Table 4). Based on the FT-IR analysis, there were no significant changes in the shape and location of the spikes in both samples. The results showed that the nanonization process did not affect LTZ’s chemical composition.

### 2.5. Carotenoids and Flavonoids Solubility in Subcritical Water

A pioneering study of degradation and solubility of fragrance and flavonoids was conducted by Yang et al., which is an investigation of the degradation of *α*-pinene, limonene, camphor, citronellol, and carvacrol, terpene member compounds, under subcritical water conditions (423 K–523 K) and their extraction from basil and oregano leaves [46] (Table 5). They reported that the stability of terpenes is highly dependent on temperature and begins to decompose as the temperature increases. In total, 25–31% of *α*-pinene and limonene were found to be degraded after 30 min of heating at 373–423 K; moreover, increasing the temperature to 523 K resulted in a 64% degradation rate of these two compounds. Nevertheless, camphor, citronellol, and carvacrol showed better stability at these temperatures, as a below 10% of degradation rate was obtained at ≤273 K and 20–42% of degradation rates were obtained at 473–523 K for these compounds.

The aqueous solubility of (+)-catechin hydrate was found between 1.32 × 10^−4^ and 3.52 × 10^−2^ between 298 and 415 K using a dynamic flow apparatus by Srinivas et al. [27]. To understand recovering flavonoid compounds from food and natural products, subcritical water can be used as a processing solvent. For this purpose, the aqueous solubilities of quercetin and quercetin dihydrate in subcritical water were investigated in a dynamic flow type apparatus by Srinivas et al. [47] using Equation (22).
(22)$$x_{s}=\frac{1}{1+\left[\frac{M_{s}}{M_{w}}\times \left(\frac{1}{S(g/L)}-1\right)\right]}$$
where *x*_s_ is the mole fraction, *M_s_* and *M_w_* are the molecular weights of the solutes and water, respectively, and *S* is the aqueous solubility of the solute in grams per liter of solvent.

The mole fraction solubility of anhydrous quercetin changed from 2.05 × 10^−7^ at 298 K to 7.12 × 10^−5^ at 413 K and that of quercetin dihydrate changed from 1.38 × 10^−7^ at 298 K to 8.58 × 10^−5^ at 413 K. The solubility of both substances increased significantly with increasing temperature. In addition, they indicated that the aqueous solubility of both molecules showed similar solubility behavior up to 353 K, but at temperatures equal to or higher than 373 K, the quercetin dehydrate form dissolved 1.5–2 times more than the anhydrous form. The solubility of the anhydrate and dihydrate forms of quercetin at different temperatures was correlated with the Apelblat equation. The importance of optimizing the solvent flow rate at a given temperature to effectively dissolve a solute such as quercetin in water was emphasized.

Furthermore, both materials were treated at different temperatures, and their morphologies were examined by scanning electron microscopy. It was determined that the particle size of quercetin dihydrate crystal decreased with increasing temperature from 353 to 413 K. This is due to the higher solubility with increasing temperature.

The solubility of *β*-carotene in subcritical water and ethanol-modified subcritical water mixture was investigated by Mottahedin et al. [48] and Ebrahimi et al. [49]. The response surface methodology was used to determine the optimal experimental conditions, and the independent variables were selected as temperature, subcritical water flow rate and % (*v*/*v*) of ethanol as co-solvent. The solubility values of *β*-carotene in both studies are given in Table 5. Although the solubility increased up to 358 K, it later decreased due to the thermal degradation of *β*-carotene. The same group conducted both studies, and there are only differences between the correlation models. In the first study, they used the cubic-plus-association equation of state (CPA EOS), and the second dielectric constant model was used to correlate the solubility of *β*-carotene in subcritical water.

The solubility of the curcumin molecule, which is biologically very active, was investigated under subcritical water conditions [50]. Experimental studies were performed under constant pressure at a temperature between 363 and 423 K using a range of 0–10% (*w*/*w*) ethanol as a co-solvent. The response surface methodology was used to determine the optimal experimental conditions, and the independent variables were selected as temperature, subcritical water flow rate and % (*v*/*v*) of ethanol as co-solvent. The maximum solubility of curcumin was found to be 230.831 ppm at a 10% ethanol–water (*v*/*v*) mixture at 423 K. The curcumin molecule shows good stability up to a temperature of 463 K. Curcumin solubility modeling was performed using the CPA equation of state in studied conditions.

Another study on the solubility of *β*-carotene under subcritical water conditions was performed by Mohammadi et al. [51]. Experimental studies were carried out under constant pressure in the temperature range of 298–403 K and using 10% by-weight ethanol as a co-solvent. The mole fraction solubility of *β*-carotene at the studied conditions was calculated in the range from 1.084 × 10^−8^ to 227.1 × 10^−8^. When ethanol is added to pure water, the solubility increases by 1.5-fold compared to that of pure water. This can be explained by the change in the dielectric constant of the mixture. The solubility of the *β*-carotene at different temperatures was correlated with the Apelblat equation. It was determined that there was no degradation in the structure of *β*-carotene at the studied temperatures, and the degradation started after 423 K.

### 2.6. Carbohydrates Solubilities in Subcritical Water

Due to the physicochemical properties of carbohydrates, they play many roles in the design and optimization of chemical engineering processes (Appendix A). The solubility of three different carbohydrates such as glucose, maltose and xylose was investigated by Zhang et al. [52] in the temperature range from 293 to 453 K. The aqueous solubility of glucose was found to vary between 4.45 × 10^−2^ at 298 K and 2.32 × 10^−1^ at 453 K. The mole fraction solubility of maltose at the same temperatures varied between 2.44 × 10^−2^ and 1.47 × 10^−1^, and xylose varied between 5.08 × 10^−2^ and 2.07 × 10^−1^. While the solubility of all three substances increased normally up to the boiling point of water, there was a five-fold increase in solubility for all molecules above the boiling point of water. Although the solubility values differ partially from the study of Yalkowsky and He [53], they are generally compatible with the literature. The solubility of the sugar was measured using the continuous flow technique, in which the sugar was saturated at various temperatures in a stream of flowing hot water. The resultant sugar solubility trends were modeled empirically or by use of a modified Apelblat equation or the modified UNIQUAC functional group activity coefficient (A-UNIFAC) model. The thermodynamic properties of the solution for the sugars and the free energies of the solution were found for all molecules to be positive and of similar magnitude.

### 2.7. Preservative Ingredient Solubilities in Subcritical Water

The solubility of parabens such as methyl, ethyl and butylparaben was measured using a homemade system in the temperature range of 273–473 K [54] (Appendix A). The mole fraction solubilities of methylparaben, ethyl and butyl were found to be 2.50 × 10^−4^, 0.74 × 10^−4^ and 0.18 × 10^−4^ at 298 K and 1.50 × 10^−3^, 0.91 × 10^−3^ and 0.41 × 10^−3^ at 473 K, respectively. The solubility of all parabens increases with increasing temperature, and the solubility increased between 6 and 19-fold by increasing the temperature from 298 to 373 K. However, severe degradation of all three parabens studied occurred at 473 K. They determined that because of the degradation of parabens at high temperature, the experimental solubility of parabens decreased. As can be seen in Equation (23), a new model was announced for the approximate calculation of paraben solubility in subcritical water. This model can reasonably predict all three parabens’ solubilities at temperatures up to 423 K.
(23)$$\ln x_{2}=\left(\frac{T_{0}}{T}\right)\ln x_{2}\left(T_{0}\right)+0.5(C-1)\left(\frac{T_{0}}{T}-1\right)$$
where *C* corresponds to the number of carbon atoms in the alkyl group.

### 2.8. Fatty Acids Solubilities in Subcritical Water

In a study on the solubility of saturated fatty acids (caprylic acid, capric acid, lauric acid, myristic acid, stearic acid and palmitic acid) with carbon numbers from 8 to 18, the effects of the temperature at 333–503 K and the pressure in the 5–15 MPa range on the solubility were investigated [55]. It was reported that the solubility of fatty acids was increased by increasing the temperature, while it did not change with the pressure. It was determined that the logarithm of the mole fraction of solubility at temperatures higher than 433 K was linearly related to the inverse of the absolute temperature for each fatty acid, and this phenomenon was associated with the formation of a regular solution of the water containing solubilized fatty acid molecules at higher temperatures. In addition, the enthalpy of a fatty acid solution in water, which was calculated by Equation (24), was reported to be increased linearly with the carbon number of the fatty acid.
(24)$$\Delta {\bar{H}}_{F}=-R{\left(\partial \ln X_{F}/\partial (1/T)\right)}_{sat}$$
where $\Delta {\bar{H}}_{F}$ indicates the enthalpy of the solution, *X*_F_ indicates mole fraction, *T* indicates reciprocal absolute temperature, and *R* indicates the gas constant.

The solubility of oleic and linoleic acid was measured in the temperature range of 333–503 K at 15 MPa by Khuwijitjaru et al. [56]. Consistent with the findings of the previous study [53], they indicated that the logarithm of the solubility of the studied fatty acids was related to the reciprocal of the absolute temperature (Equation (24)). In addition, the enthalpy of the solution of both oleic and linoleic acid in the liquid state in water was reported to be 122 kJ/mol.

The decompositions of monocaprylin, monocaprin, monolaurin and their corresponding fatty acids (caprylic, capric and lauric acids) under subcritical water conditions were investigated by Fujii et al. [57]. It was found that monoacyly glycerol was hydrolyzed to the corresponding fatty acid and glycerol, where fatty acid was further decomposed. It was assumed that the degradation of monocaprylin under isothermal conditions in subcritical water obeys first-order kinetics, and the decompositions of caprylic, capric and lauric acids were expressed with the same kinetics under the same conditions. The Arrhenius equation (Equation (25)) was employed to express the temperature dependence of the rate constant (*k_i_*) to determine the decomposition of monoacyly glycerol and fatty acids.
(25)$$k_{i}=k_{i0}\exp(-E_{i}/RT)$$
where *k_i_*_0_, *E_i_*, and *R* indicate the frequency factor, the activation energy and the gas constant, respectively.

Huang et al. [58] studied the solubility of fatty acids under critical conditions. These two fatty acids have different polarities and partially stable thermal properties. This is due to the different carbon numbers in the aliphatic chain. They designed a phase equilibrium device to determine the solubility of stearic and palmitic acid in subcritical water and measured the solubility of these substances at different temperatures and pressures. Furthermore, the effect of the ultrasonic field on dissolution equilibrium was investigated. The maximum solubilities of stearic acid and palmitic acid were found to be 0.136 g/100 g and 0.178 g/100 g at temperatures of 180 and 160 °C, respectively, and a pressure of 15 MPa for 30 min in subcritical water. It has been determined that the ultrasonic output frequency and power affect the dissolution, and the dissolution is better at low frequency and high power because lower frequencies resulted in stronger ultrasonic cavitation and reduced acoustic attenuation [59]. In addition, all studies about the solubility of fatty acids are given in Table 6.

### 2.9. Pesticide Solubilities in Subcritical Water

To understand and improve water cleaning using subcritical water, a few studies were conducted by several scientific groups (Table 7). Miller et al. [11] studied some pesticide solubilities (propazine, chlorothalonil and endosulfan II) in subcritical water, where they observed a quite high increase in solubility by increasing the temperature.

The solubility of three triazine pesticides, which are known as food contaminants, namely atrazine, cyanazine, and simazine, were measured in pure and modified subcritical water at a temperature range of 323–373 °C and a pressure of 50.6 bar using Equation (26) [60]. It was indicated that the temperature and the amount and type of the modifier (cosolvent) are major system variables that influence the solvent strength, as the dielectric constant of water is highly dependent on these variables. Therefore, ethanol and urea were used in a variable amount and variable temperature range to modify the water. Herein, it was determined that an increase in each 25 K in temperature provides a three-fold increase in the solubility of triazine. In addition, when water is modified with urea and ethanol, respectively, the solubility of atrazine is obtained to be doubled and increased over an order of magnitude. Atrazine was reported to be increased by 25-fold as the temperature was raised from 323 to 373 K (Table 7).
(26)$$\ln\left[x_{1}(T)\right]\approx \left(\frac{T_{0}}{T}\right)\ln\left[x_{1}(T_{0})\right]+15{\left[(\frac{T}{293})-1\right]}^{3}$$
where *x*_1_ indicates mole fraction solubility, and *T*_0_ is the temperature at which the solute is close to the medium in which the water solubility of the compound is known.

## 3. Conclusions

The development of sustainable technologies in a world with a growing population and limited resources is of great interest in the 21st century. Therefore, it is very important to use environmentally friendly and sustainable materials in as many fields as possible in today’s world. Subcritical water is one of the most important resources that can be used for this purpose. Understanding the solubility and stability of organic substances in subcritical water has far broader implications in many fundamental and applied areas, including environmental remediation, analytical-scale extractions with subcritical water, and subcritical water chromatography. For example, if certain compounds are stable in subcritical water at a certain temperature, water at or below that temperature can be used in extraction and chromatographic separations of these compounds for quantification. In contrast, if pollutants decompose under subcritical water conditions, subcritical water can be used to treat or remediate contaminated environmental solids, such as soils, sediments, and sludges. It is clear that organic solubilities increase with higher water temperatures. In general, the degree of degradation of organic compounds also increases with increasing water temperature. To further develop green subcritical water separation techniques, fundamental data such as solubility and decomposition of more organic compounds are critically needed. Although there are solubility models that have become available over the last couple of decades to predict organic solubility in subcritical water, more accurate and universal models still need to be developed. For environmental remediation purposes, catalysts or other reagents need to be investigated to increase the destruction efficiency of toxic pollutants or to convert toxic pollutants to nontoxic species. Lastly, certain chemical syntheses may be carried out in subcritical water media.

## Figures and Tables

**Figure 1 molecules-28-01000-f001:**
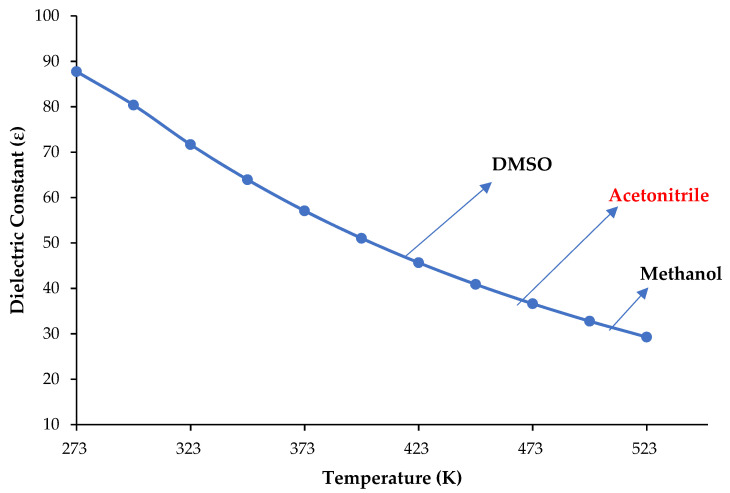
The dielectric constant of water changes with the temperature at 50 bar (data taken from Refs. [1,2,3]).

**Figure 2 molecules-28-01000-f002:**
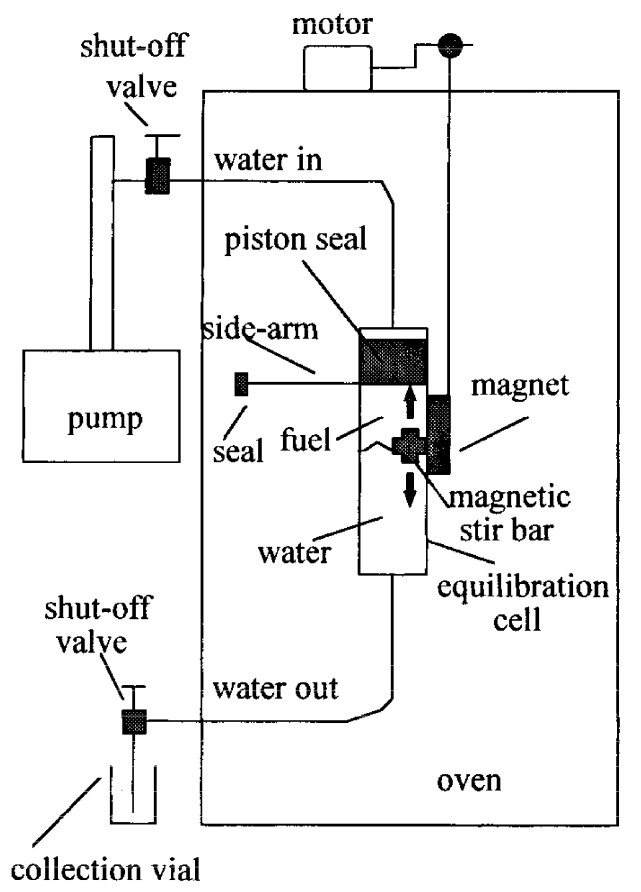
Experimental system of determination solubility and partitioning behavior of fuel components in subcritical water designed by Yang et al. (Adopted from Reference [9] with permission from the American Chemical Society; Copyright 1997).

**Table 1 molecules-28-01000-t001:** The solubilities of polycyclic aromatics and derivatives in subcritical water.

Compound	Temperature Range (K)	Pressure (Bar)	Solvent	Solubility Range (x_2_) ^a^	Empirical Equations	Reference
Anthracene	298–498	30–60	Subcritical (superheated) water	8.10 × 10^−9^ to 2.10 × 10^−4^	$\ln x_{2}(T)\approx (\frac{T_{0}}{T})\ln x_{2}(T_{0})$	[11]
Pyrene				1.07 × 10^−8^ to 1.00 × 10^−7^		
Chrysene				6.30 × 10^−10^ to 7.58 × 10^−5^		
Perylene				2.90 × 10^−10^ to 5.00 × 10^−6^		
Carbazole				1.10 × 10^−7^ to 1.90 × 10^−3^		
Naphthalene	298–473	30–70	Subcritical (superheated) water	4.50 × 10^−6^ to 3.04 × 10^−5^	$\ln x_{2}(T)=(\frac{T_{0}}{T})\ln x_{2}(T_{0})+15{\left(\frac{T}{T_{0}}-1\right)}^{3}$	[12]
Benzo[a]pyrene				2.90 × 10^−10^ to 7.82 × 10^−5^		
Acenaphthene	323–573	50–100	Subcritical (superheated) water	1.25 × 10^−3^	N/A	[13]
Anthracene				1.02 × 10^−7^–3.78 × 10^−3^		
Pyrene				6.87 × 10^−8^–1.41 × 10^−3^		
Naphthalene	298–498	1–70	Subcritical (superheated) water	4.50 × 10^−6^–4.35 × 10^−5^	$x_{2}=\frac{f_{2}^{s0}}{\gamma_{2}^{sat}f_{2}^{10}}$ $\ln\frac{f_{2}^{s0}}{f_{2}^{10}}\approx \frac{\Delta h_{2}^{\mathrm{fus}}}{RT_{t2}}\left(1-\frac{T_{t2}}{T}\right)+\frac{\left(v_{2}^{s0}-v_{2}^{10}\right)P}{RT}$	[14]
Anthracene				7.40 × 10^−9^–2.20 × 10^−4^		
Pyrene				1.07 × 10^−8^–5.40 × 10^−6^		
Chrysene				1.60 × 10^−10^–7.58 × 10^−5^		
1,2-benzanthracene				3.37 × 10^−9^–2.96 × 10^−6^		
Triphenylene				1.82 × 10^−9^–2.83 × 10^−5^		
Perylene				3.00 × 10^−11^–5.00 × 10^−6^		
*p*-terphenyl				8.49 × 10^−10^–3.93 × 10^−5^		
Naphthalene	313–483	40–80	Subcritical (superheated) water	6.92 × 10^−6^–4.35 × 10^−5^	$\ln x_{2}=a_{1}+a_{2}\left(\frac{T_{0}}{T}\right)+a_{3}\ln(\frac{T}{T_{0}})$	[15]
Anthracene				1.19 × 10^−8^–2.20 ×10^−3^		
1,2-benzanthracene				3.37 × 10^−9^–2.96 × 10^−6^		
Triphenylene				1.82 × 10^−9^–2.83 × 10^−5^		
*p*-terphenyl				0.849 × 10^−9^–3.93 × 10^−5^		
Phenanthrene	313–453	50	Subcritical (superheated) water	2.17 × 10^−7^ to 3.27 × 10^−6^	$\ln x_{2}=b_{1}+b_{2}\left(\frac{T_{0}}{T}\right)+b_{3}\ln(\frac{T}{T_{0}})$	[16]
Phenanthridine				6.29 × 10^−6^ to 5.92 × 10^−5^		
Acridine				9.10 × 10^−6^ to 6.09 × 10^−5^		
Phenazine				7.17 × 10^−6^ to 8.27 × 10^−4^		
Thianthrene				1.50 × 10^−8^–1.61 × 10^−5^		
Phenothiazine				2.92 × 10^−7^–4.31 × 10^−4^		
Phenoxathiine				2.28 × 10^−7^–7.51 × 10^−7^		
Phenoxazine				1.94 × 10^−6^ – 2.23 × 10^−4^		
Carbazole				2.72 × 10^−7^–1.68 × 10^−4^		
Dibenzofuran				9.17 × 10^−7^–7.04 × 10^−6^		
Dibenzothiophene				2.06 × 10^−7^–3.49 × 10^−6^		
4,6-DMDBT				3.17 × 10^−9^–5.15 × 10^−6^		
Anthracene, Perylene, Benzo-pyrene, Pyrene, Chrysene, Naphthalene, Fluorene, Fluoranthene, Phenanthrene, 1,2-Benzanthracene, *p*-terphenyl	298–500	N/A	Subcritical water (pressurized hot water)	N/A	$\ln x_{2}^{\mathrm{id}}=\ln\left(\frac{f_{2}^{s}}{f_{2}^{o}}\right)=-\frac{\Delta H_{m2}}{RT_{m2}}\left(\frac{T_{m2}}{T}-1\right)$ $\ln x_{2}=\ln x_{2}^{\mathrm{id}}-\ln\gamma_{2}$ UNIFAC-based thermodynamic models	[17]
2-methylanthracene	313–513	50–64	Pressurized hot water	5.23 × 10^−9^–3.06 × 10^−5^	$\ln x_{2}=a_{1}+a_{2}\left(\frac{T_{0}}{T}\right)+a_{3}\ln(\frac{T}{T_{0}})$	[18]
9,10-dimethylanthracene				3.27 × 10^−9^–1.24 × 10^−5^		
9-phenylanthracene				7.57 × 10^−10^–1.2 × 10^−6^		
Triptycene				2.69 × 10^−10^–5.39 × 10^−4^		
Fluorene, Biphenyl, Triphenylene, Benz[a]anthracene, Naphthalene, Anthracene, Pyrene, Fluoranthene, Chrysene, Acenaphthene, and Phenanthrene	313–498	40–65	Subcritical water (pressurized hot water)	N/A	$x_{s}=\frac{\varphi_{s}^{L0}}{\varphi_{s}^{L}}\exp\left[-\frac{\Delta_{f\mathrm{us}}H}{R}\left(\frac{1}{T}-\frac{1}{T_{m}}\right)\right]$	[19]
Xanthene	313–473	50	Pressurized hot water	2.52 × 10^−7^–5.56 × 10^−6^	$\ln x_{2}=a_{1}+a_{2}\left(\frac{T_{0}}{T}\right)+a_{3}\ln(\frac{T}{T_{0}})$	[20]
Anthrone				3.45 × 10^−7^–1.26 × 10^−4^		
Xanthone				7.09 × 10^−7^–2.71 × 10^−4^		
Thioxanthone				1.18 × 10^−7^ to 3.83 × 10^−4^		
9,10-anthraquinone				7.25 × 10^−8^ to 2.96 × 10^−5^		
9,10-phenanthrenequinone				5.50 × 10^−7^ to 1.83 × 10^−3^		
Anthracene	393–443	50–150	Subcritical water	1.22 × 10^−6^–2.84 × 10^−5^	The Peng−Robinson equation of state (PR-EOS)	[21]
*p*-terphenyl				1.82 × 10^−7^–8.67 × 10^−6^		
Anthracene	393–443	50–150	Subcritical ethanol	1.64 × 10^−2^–6.82 × 10^−2^	UNIQUAC, O-UNIFAC, and M-UNIFAC models	[23]
			f = 0.10 ethanol-modified subcritical water	9.09 × 10^−5^–1.11 × 10^−3^		
*p*-terphenyl			Subcritical ethanol	6.55 × 10^−3^–9.54 × 10^−2^		
			f = 0.10 ethanol-modified subcritical water	1.34 × 10^−5^–2.26 × 10^−4^ to		

^a^ Mole fraction solubility of compounds; N/A: not available.

**Table 2 molecules-28-01000-t002:** Alkyl and Chlorobenzene solubilities in subcritical water.

Compound	Temperature Range (K)	Pressure (Bar)	Solvent	Solubility Range (x_2_) ^a^	Empirical Equations	Reference
Ethylbenzene	298–473	50	High-temperature water (subcritical water)	2.80 × 10^−5^ to 8.10 × 10^−4^	$\ln x_{2}(T)=(\frac{T_{0}}{T})\ln x_{2}(T_{0})+2{\left(\frac{T-T_{0}}{T_{0}}-1\right)}^{3}$	[24]
*m*-xylene				3.70 × 10^−5^ to 1.02 × 10^−3^		
Benzene				4.20 × 10^−4^ to 4.60 × 10^−3^		
Chlorobenzene	446–540	22	Subcritical water	6.90 × 10^−4^–1.13 × 10^−3^	N/A	[25]
4-chlorotoluene	535–566	50–500	Subcritical water	4.59 × 10^−3^–1.36 × 10^−2^	N/A	[26]

^a^ Mole fraction solubility of compounds; N/A: not available.

**Table 3 molecules-28-01000-t003:** Organic acid solubilities in subcritical water.

Compound	Temperature Range (K)	Pressure (Bar)	Solvent	Solubility Range (*x*_2_) ^a^	Empirical Equations	Reference
Benzoic acid	298–473	50	Subcritical water	2.22–1.36 × 10^2^	$\ln x_{2}(T)=(1.85\frac{T_{0}}{T}-1)\ln x_{2}(T_{0})$	[27]
Salicylic acid				4.69 × 10^−1^–1.02 × 10^2^		
Gallic acid	298–415	3.5	Subcritical water	1.24 × 10^−3^–2.33 × 10^−1^	$\ln x_{2}=\left(\frac{T_{0}}{T}\right)\ln x_{2}\left(T_{0}\right)+11\left(1-\frac{T_{0}}{T}\right)$	[28]
Protocatechuic acid				3.55 × 10^−3^–1.26 × 10^−1^		
Terephthalic acid	349–547	100	Subcritical water	1.25 × 10^−5^ to 2.99 × 10^−2^	N/A	[29]
Sebacic acid	313–433	50	Subcritical water	2.22 × 10^−5^ to 25.69 × 10^−3^	$\ln x_{2}(T)=(1.85\frac{T_{0}}{T}-1)\ln x_{2}(T_{0})$	[30]

^a^ Mole fraction solubility of compounds; N/A: not available.

**Table 4 molecules-28-01000-t004:** Pharmaceutical solubilities in subcritical water.

Compound	Temperature Range (K)	Pressure (Bar)	Solvent	Solubility Range (*x*_2_) ^a^	Empirical Equations	Reference
Antiviral agentvitamin B3 flavonoids	323–398	N/A	Subcritical water and compressed hydroethanolic mixtures	N/A	$R_{a}^{2}=4{\left(\delta_{D_{1}}-\delta_{D_{2}}\right)}^{2}+{\left(\delta_{P_{1}}-\delta_{P_{2}}\right)}^{2}+{\left(\delta_{H_{1}}-\delta_{H_{2}}\right)}^{2}$	[31]
Budesonide	373 to 433	70	Subcritical water	8.31 × 10^−7^–4.53 × 10^−5^	The M-UNIFAC and MF-UNIFAC models	[32]
Antifungal drug–griseofulvin	413 to 443	70	Subcritical water	1.60 × 10^−4^–5.28 × 10^−4^	M-UNIFAC and MF-UNIFAC model	[33]
Naproxen	403 to 443	70	Subcritical water	4.09 × 10^−6^–5.56 × 10^−5^	M-UNIFAC model	[34]
Paracetamol	293 to 403	50	Subcritical water	1.52 × 10^−3^–1.47 × 10^−2^	$\ln x_{2}=\left(\frac{T_{0}}{T}\right)\ln x_{2}\left(T_{0}\right)+\frac{8}{5}\left(\frac{T-T_{0}}{T_{0}}\right)$	[35]
Anticancer drug– 5-Fluorouracil	298 to 473	51	Subcritical water	1.69 × 10^−3^–2.10 × 10^-−2^	$\;\ln x_{2}=\left(\frac{T_{0}}{T}\right)\ln x_{2}\left(T_{0}\right)+\frac{\varepsilon_{\left(T\right)}}{\varepsilon_{\left(T_{0}\right)}}{\left[\left(\frac{T-T_{0}}{T_{0}}\right)\right]}^{2}$	[36]
Antibiotic drug–ciprofloxacin	373 to 443	40	Subcritical water–ethanol mixture	2.0 × 10^−6^–5.50 × 10^−5^	The modified Apelblat model	[37]
Bicalutamide	383 to 443	55	Subcritical water	7.90 × 10^−6^–6.24 × 10^−4^	Modified Apelblat model	[38]
Megestrol acetate				2.70 × 10^−6^–9.90 × 10^−5^		
Prednisolone				1.63 × 10^−4^–58.70 × 10^−4^		
Clarithromycin				3.56 × 10^−4^–22.81 × 10^−4^		
Beclomethasone dipropionate				2.00 × 10^−6^–2.13 × 10^−5^		
Antibiotic drug–sulfadiazine	343 to 403	51	Subcritical water	0.57 × 10^−4^–6.30 × 10^−4^	Modified Apelblat equation and CNIBS/R-K model	[39]
			20% ethanol-modified subcritical water	0.23 × 10^−3^–2.09 × 10^−3^		
			20% propanol-modified subcritical water	0.27 × 10^−3^–3.24 × 10^−3^		
Antidepressant drug– escitalopram	298 to 473	50	Subcritical water	2.94 × 10^−3^ to 60.83 × 10^−3^	lnx2=(T0T)lnx2(T0)+(ε(T0)ε(T))12	[40]
Antifungal drug–fluconazole	298 to 473	50	Subcritical water	2.40 × 10^−4^–11.31 × 10^−3^	$\ln x_{2}\left(T\right)=\left(\frac{T_{0}}{T}\right)\ln x_{2\;}\left(T_{0}\right)+\ln\left(\frac{\varepsilon_{T_{0}}}{\varepsilon_{T}}+2\right)$	[41]
Ibuprofen	298 to 473	50	Subcritical water	2.30 × 10^−4^ to 21.36 × 10^−2^	lnx2(T)=(1.85T0T−1)lnx2(T0)+(ε(T)ε(T0))TT0	[42]
Ampicillin	303 to 403	50	Subcritical water	3.80 × 10^−4^–17.69 × 10^−3^	N/A	[43]
Antiarrhythmic drug–amiodarone hydrochloride	298 to 393	50	Subcritical water	0.14 × 10^−4^ to 2.85 × 10^−4^	The linear and modified Apelblat models	[44]
			5% ethanol-modified subcritical water	0.31 × 10^−4^ to 5.58 × 10^−4^		
			10% propanol-modified subcritical water	0.50 × 10^−4^–9.81 × 10^−4^		
Anticancer drug–letrozole	298 to 383	50	Subcritical water	5.00 × 10^−6^ to 1.16 × 10^−4^	The linear and the Apelblat models	[45]

^a^ Mole fraction solubility of compounds; N/A: not available

**Table 5 molecules-28-01000-t005:** Carotenoid and flavonoid solubility and degradation in subcritical water.

Compound	Temperature Range (K)	Pressure (Bar)	Solvent	Solubility Range (*x*_2_)^a^	Empirical Equations	Reference
*α*-pinene, Limonene, Camphor, Citronellol, and Carvacrol	423–523	16	Subcritical Water	Degradation Range 20–42%	N/A	[46]
Catechin hydrate	298–415	3.5	Subcritical water	1.32 × 10^−4^–3.52 × 10^−2^	$\ln x_{2}=\left(\frac{T_{0}}{T}\right)\ln x_{2}\left(T_{0}\right)+11\left(1-\frac{T_{0}}{T}\right)$	[28]
Quercetin	298–413	N/A	Subcritical water	2.05 × 10^−7^–7.12 × 10^−5^	Modified Apelblat equation	[47]
Quercetin dihydrate				1.38 × 10^−7^–8.58 × 10^−5^		
*β*–carotene	343–403	50	Subcritical water	1.08 × 10^−8^ –1.20 × 10^−6^	The cubic-plus-association equation of state (CPA EOS)	[48]
			5% ethanol-modified subcritical water	1.34 × 10^−8^–2.04 × 10^−6^		
			10% ethanol-modified subcritical water	1.52 × 10^−8^–2.27 × 10^−6^		
*β*–carotene	343–403	20	Pressurized hot water	3.53 × 10^−6^–6.04 × 10^−6^ β–carotene decomposed after 403 K	Dielectric constant model ln x = Ad + B	[49]
Curcumin	363–423	50	Subcritical water	1.08 × 10^−8^–1.20 × 10^−6^	The linear model and modified Apelblat model	[50]
			5% ethanol-modified subcritical water	1.34 × 10^−8^–20.45 × 10^−7^		
			10% ethanol-modified subcritical water	1.52 × 10^−8^–22.71 × 10^−7^		
*β*–carotene	298–403	20	Subcritical water	9.47 × 10^−4^–2.29 × 10^−3^	Cubic-plus-association equation of state (CPA EOS) and Dielectric constant model	[51]
			5% ethanol-modified subcritical water	1.37 × 10^−3^–5.69 × 10^−3^		
			10% ethanol-modified subcritical water	2.31 × 10^−3^–7.27 × 10^−3^		

^a^ Mole fraction solubility of compounds; N/A: Not Available.

**Table 6 molecules-28-01000-t006:** Fatty acid solubilities in subcritical water.

Compound	Temperature Range (K)	Pressure (Bar)	Solvent	Solubility Range (*x*_2_) ^a^	Empirical Equations	Reference
Caprylic acid	333–573	50–150	Subcritical water	≈1.8 × 10^−7^ to 1.8 × 10^−3^	$\Delta {\bar{H}}_{F}=-R{\left(\partial \ln X_{F}/\partial (1/T)\right)}_{sat}$	[54]
Capric acid						
Lauric acid						
Myristic acid				≈9.9 × 10^−8^ to 1.8 × 10^−2^		
Stearic acid						
Palmitic acid						
Oleic	333–503	150	Subcritical water	≈10^−8^–10^−2^	$\Delta {\bar{H}}_{F}=-R{\left(\partial \ln X_{F}/\partial (1/T)\right)}_{sat}$	[55]
Linoleic						
Stearic acid	433–453	150	Subcritical water	0.0–8.61 × 10^−5^	N/A	[57]
Palmitic acid				0.0–1.25 × 10^−5^		

^a^ the mole fraction solubility of compounds; N/A: not available.

**Table 7 molecules-28-01000-t007:** Pesticide solubilities in subcritical water.

Compound	Temperature Range (K)	Pressure (Bar)	Solvent	Solubility Range (*x*_2_) ^a^	Empirical Equations	Reference
Propazine,	298–498	30–60	Subcritical water	4.93 × 10^−7^–2.10 × 10^−3^	$\ln x_{2}(T)\approx (\frac{T_{0}}{T})\ln x_{2}(T_{0})$	[11]
Chlorothalonil				1.22 × 10^−3^–1.58 × 10^−2^		
Endosufan II				1.19 × 10^−7^–1.99 × 10^−4^		
Simazine	323–373	50.6	Modified Subcritical water	1.5 × 10^−6^ to 2.1 × 10^−5^	$\ln\left[x_{1}(T)\right]\approx \left(\frac{T_{0}}{T}\right)\ln\left[x_{1}(T_{0})\right]+15{\left[(\frac{T}{293})-1\right]}^{3}$	[60]
Atrazine				5.8 × 10^−6^ to 4.6 × 10^−5^		
Cyanazine				3.6 × 10^−5^ to 1.5 × 10^−4^		

^a^ Mole fraction solubility of compounds.

## Data Availability

Not applicable.

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
