# Peer review of "Solubility and Decomposition of Organic Compounds in Subcritical Water"

_molecules, 2023, doi:10.3390/molecules28031000_

Round 1
Reviewer 1 Report
The review discusses the main experimental data concerning the solubility of organic compounds as well their stability under subcritical water conditions, which is very important for many domains, a part of them being included in this material. The manuscript is very well written and detailed, it could be published in this journal, only a few comments are added to its content, which should be included in a new version before its acceptance for publications.
1. An important consequence of the solubility of organic compounds refers to the influence of subcritical water conditions upon the various molecular interactions between organic compounds and different adsorbents, mainly used in several separation techniques. This aspect could be included even shortly in this review with several references in order to complete this useful domain. I give here two examples of studies that have in their objective the use of subcritical water, which do not belong to the referee:
i) J. Chromatogr. A, 1998, 810 (1-2), 149-159.
ii) TrAC Trends in Analytical Chemistry, 2020, 123, 115793.
2) A quite recent review covering a similar topic is recommended to be included, published in Brazilian Journal of Chemical Engineering (Vol. 36, No. 04, pp. 1367 - 1392, October - December, 2019 dx.doi.org/10.1590/0104-6632.20190364s20170493).
Reviewer 2 Report
The manuscript presents data on the solubility and decomposition of organic compounds in subcritical water. The subject addressed by the manuscript is current and provides theoretical and practical data regarding the behavior of some chemical compounds in subcritical water at certain temperature and pressure values.
1. The manuscript contains figures taken from the works of some authors. Is there the agreement of these authors/the journal in which the article appears for taking these figures? If yes, this consent must be specified under each figure.
2. There are many data presented both in the text and in the tables. The data should be reorganized.
3. There are some spelling errors:
Line 40
Figure 1 . Acetonitril
Line 75
Line 128
Line 144
Line 166
Reviewer 3 Report
Solubility and Decomposition of Organic Compounds in Subcritical Water
molecules-2140167
Reviewer’s Comments
The authors have presented review about Solubility and Decomposition of Organic Compounds in Subcritical Water. However, some issues need addressing before the manuscript is suitable for publication.
1. 40 – It should be “constant.”.
2. 72 - It should be “Miller et al.”.
3. 75 – It should be “equation based”.
4. 80 – It should be “Where x2(T0)” and and there should be a comma at the end of the line.
5. 81 - There should be a period at the end of the line.
6. 194-197 - Different line spacing than for the whole text.
7. 245 - This verse should start with "Where".
8. 258 - It should be “ethylbenzene, m-xylene”.
9. 313 - It should be “Where”.
10. 365 - It should be “Multiple”.
11. 383 - It should be “Carr et al.”.
12. 422 - It should be “the stabilities”.
13. 532 - It should be “Where”.
14. 607 - It should be “1.5x10-3 at”.
15. 625 - It should be “the”.
16. 694 - The equation should be in the middle.
17. The font size for equations varies. Please standardize as much as possible.
18. Literature references should be given in one scheme. There are many schemes at work. Please unify it.
Reviewer 4 Report
lines 126-130 and the furthers: unify the PAHS vs. PAHs
line 288 for which compound is degraded salicylic acid?
table 2 - change the format
line 607 methylparaben - together, ethyl and butyl too...
table 8 - names of acid should be started with capitals. the same is for table 9
in conclusion, some future trends and perspectives should be mentioned
Round 2
Reviewer 2 Report
The authors made the appropriate changes to improve the quality of the manuscript. Tables 6 and 7, in my opinion, reproduce the comments in the text, therefore I suggest to bring additional data (if any) or to give them up.
Author Response
The manuscript was revised and Table 6 and Table 7 were removed from the manuscript to the Supplementary materials file.